# Rapid Response to the Combination of Lenvatinib and Pembrolizumab in Patients with Advanced Carcinomas (Lung Adenocarcinoma and Malignant Pleural Mesothelioma)

**DOI:** 10.3390/cancers13143630

**Published:** 2021-07-20

**Authors:** Walid Shalata, Muhammed Iraqi, Baisali Bhattacharya, Vered Fuchs, Laila C. Roisman, Ahron Yehonatan Cohen, Ismaell Massalha, Alexander Yakobson, Manu Prasad, Moshe Elkabets, Angel Porgador, Nir Peled

**Affiliations:** 1The Legacy Heritage Center & Dr. Larry Norton Institute, Soroka Medical Center, Beer Sheva 84105, Israel; walid.shalata@gmail.com (W.S.); aharonc@clalit.org.il (A.Y.C.); ismaell@post.bgu.ac.il (I.M.); alexy@clalit.org.il (A.Y.); 2The Shraga Segal Department of Microbiology, Immunology, and Genetics, Faculty of Health Science, Ben-Gurion University of the Negev, Beer Sheva 8410501, Israel; iraqi@post.bgu.ac.il (M.I.); baisali@post.bgu.ac.il (B.B.); veredfu@post.bgu.ac.il (V.F.); manupras@post.bgu.ac.il (M.P.); moshee@bgu.ac.il (M.E.); 3Shaare Zedek Medical Center, Oncology Division and Cancer Institute, Jerusalem 9103102, Israel; Lailaro@szmc.org.il

**Keywords:** lung adenocarcinoma, NSCLC, malignant pleural mesothelioma, multikinase inhibitor, immune checkpoint inhibitor, lenvatinib, pembrolizumab

## Abstract

**Simple Summary:**

Over the past decades, the multi-drug approach for treating various types of cancers was highly recommended. The aim of this study was to demonstrate the advantage of treating different carcinoma types with several combinations of immune checkpoint inhibitors (ICIs) and tyrosine kinase inhibitors (TKIs). We showed that patients treated with the combination of lenvatinib and pembrolizumab showed an enhanced response to therapy compared to other treatments. Our ex-vivo studies exhibited reduced proliferation and PD-L1 levels for PDX explants treated with this combination compared to untreated explants or those treated with a single treatment. Based on the results of this study, we propose that the combination of levatinib and pembrolizumab shows great potential as a treatment option for patients with non-small cell lung carcinoma and malignant pleural mesothelioma, as previously suggested by recent clinical trials.

**Abstract:**

The new era of cancer treatments has made immune checkpoint inhibitors (ICIs) and emerging multikinase inhibitors (TKIs) the standards of care, thus drastically improving patient prognoses. Pembrolizumab is an anti-programmed cell death-1 antibody drug, and lenvatinib is a TKI with preferential antiangiogenic activity. We present, to our knowledge, the first reported series of cases consisting of patients with metastatic non–small cell lung cancer and malignant pleural mesothelioma who were treated with several types of chemotherapy combinations and ICIs followed by disease progression. They were subsequently treated with combined immunotherapy and TKI treatment, resulting in a near complete response within a very short time. Clinical responses were supported by in vitro testing of each patient’s lymphocytic response to pembrolizumab after pre-exposure of target cancer cells to lenvatinib.

## 1. Introduction

Lung cancer is the leading cause of cancer-related deaths in the United States and is a significant health care concern throughout the world [1]. Non–small cell lung cancer (NSCLC) accounts for approximately 85% of all lung cancers. The natural history of NSCLC is often insidious, with few, if any, symptoms until the disease is relatively widespread. Thus, most lung carcinomas are diagnosed at an advanced stage, resulting in a poor prognosis [2]. Malignant pleural mesothelioma (MPM) is one of the oncologic pathologies expected to be on the rise in the upcoming 10–20 years, secondary to asbestos exposure in the mid to late 20th century [3]. This diagnosis is also accompanied by a poor prognosis [4]. Even with current treatment schemes, overall and median survival rates are low. Mutational cues and patient background characteristics have little influence in directing therapy and predicting outcomes [5].

Pembrolizumab is a humanized IgG4 monoclonal antibody directed against the programmed cell death receptor -1 (PD-1), a major immune checkpoint receptor that regulates T-cell response. Pembrolizumab blocks PD-1 activity, thereby enhancing antitumor T-cell activity [6]. Lenvatinib is a multitargeted tyrosine kinase inhibitor of vascular endothelial growth factor receptors 1, 2, and 3, fibroblast growth factor receptors 1–4, platelet-derived growth factor receptor α, rearranged during transfection (RET and tyrosine-protein kinase (KIT) [7].

In a phase II trial, the combination of pembrolizumab and lenvatinib achieved encouraging antitumor results with a manageable safety profile in patients with selected advanced solid tumors including NSCLC [8]. The decision to combine these agents was based on the antitumor activity shown in preclinical data [9].

To demonstrate the potential efficacy of the combination of immune checkpoint inhibitors and multikinase inhibitors, we report a series of five cases, four of metastatic NSCLC and one of MPM, that responded significantly to the combination treatment of pembrolizumab and lenvatinib who were treated under local IRB approval, after progression following prior lines of therapy. To support the clinical outcomes observed, we conducted in vitro laboratory tests for four of the five patients, in order to test the CD8+ T cells’ sensitivity to pembrolizumab after their exposure to lenvatinib. This test was obtained by measurement of the interferon-gamma (IFN-γ) secretion of CD8+ T cells using the A549 cell line (carcinoma cell line) as target cells [10]. This technique was first used at Ben-Gurion University in collaboration with the Weizmann Institute of Science, using a syngeneic tumor target and CD8+ T cells [10].

## 2. Materials and Methods

### 2.1. Patient Selection

All five patients have progressed on accepted standard of care systemic therapies and were not eligible for clinical trials. Consequently, they received pembrolizumab with lenvatinib under local IRB approval as off-label systemic therapy, outside the auspices of the clinical trial (Helsinki code of ethics 0384-18 and 0005-19).

### 2.2. Isolation and Culturing of the Patients’ Peripheral CD8+ T Cells

Isolation of peripheral blood mononuclear cells (PBMCs) from the peripheral blood of the patients was carried out according to a standard protocol using lymphocyte separation medium (MP Biomedicals, Irvine, CA, SKU 0850494-CF). Then, PBMCs were cultured with 200U IL-2, and two days later, CD8+ T cells were isolated by using human CD8 Microbeads (Miltenyi Biotec, Bergisch Gladbach, North Rhine-Westphalia, 130-045-201, Bergisch Gladbach, Germany), LS column (Miltenyi Biotec, Bergisch Gladbach, North Rhine-Westphalia, 130-042-401), and MidiMACS Separator (130-042-302). RPMI containing 10% human male AB plasma (Sigma-Aldrich, MO, USA, H4522), 1 mM sodium pyruvate, 2 mM L-glutamine, 0.1 mM MEM nonessential amino acids, 1% penicillin/streptomycin, 10 mM HEPES (Life Technologies, Waltham, MA, USA), 200 IU/mL recombinant human IL-2 (200-02-500UG, PeproTech, Cranbury, NJ, USA), and 50 ng/mL antihuman CD3 Antibody (BioLegend, San Diego, CA, USA, 317302) were used for the first 48 h of culture. Then, culturing and passaging was done using complete media without antihuman CD3 antibody.

### 2.3. Patient-Derived Xenograft (PDX) Generation and Propagation

Patient-derived tumor tissue was implanted in male nonobese diabetic/severe combined immunodeficiency (NOD/SCID) subcutaneously as we have described previously [11]. Tumor implantation growth rates varied from one to six months. Generation and propagation of the PDXs were done under the guidelines of the institutional animal care and use committee of Ben-Gurion University of the Negev (IL-29-05-2018(E)).

### 2.4. Co-Culture of A549 CD8+ T Cells

At 24 h before the experiment, A549 cells were seeded using normal complete DMEM (Dulbecco’s modified eagle medium), 10% FBS (fetal bovine serum) in a 96 flat-bottom well plate with 60% confluency. The cells were washed and incubated with either lenvatinib (5 μM) in 200 μL of media, or control media for 10 h, and 10 h later, cells were washed twice with 1X PBS; CD8+ T cells at a concentration of 100,000 cells/well were added to the washed wells along with pembrolizumab (20 μg/mL) or mock, in complete RPMI media containing 20 U IL-2 and 10% human serum. After 18 h of incubation, the supernatant was collected from the wells and then assayed with a standard IFN-γ ELISA assay (ELISA MAX, Biolegend, San Diego, CA, USA).

### 2.5. Immuno-Tumor Ex Vivo Analysis (i-TEVA)

Tumors from PDX mice were collected once they reached approximately 500 mm^3^. Then, tumors were cut into 2 × 2 × 2 mm^3^ tissue explants using a unique cutting tool. All explants were cultured in 48-well tissue culture plates using DMEM culture media (TEVA media) [11]. Explants were incubated either with lenvatinib or TEVA media for 10 h, and then were washed three times with PBS 1× and incubated for 18 h with 2 × 10^5^ T cells in complete RPMI media containing 20 U of IL-2 (200-02-500UG, PeproTech, Cranbury, NJ, USA).

### 2.6. Tumor Tissue Explants Culture, Preparation of Formalin-Fixed Paraffin-Embedded (FFPE) Blocks, and Tissue Microarray

The PDXs were used for preparation of 2 × 2 × 2 mm^3^ ex vivo tumor tissue explants culture according to a previously published protocol [11]. FFPE blocks were then prepared from the explants using an automated tissue processing machine (Leica Biosystems, Nußloch, Germany) [11,12,13]. Finally, tissue microarray (TMA) blocks were prepared from donor FFPE blocks using 3-mm T-Sue^TM^ punch needles (Simport, Beloeil, QC, Canada), with each block containing up to 24 tissue explants.

### 2.7. Immunohistochemistry Staining and Quantification

Immunohistochemical staining was carried out as previously described [11] (Ki67 [1:250, Vector Laboratories, Burlingame, CA, cat no-VP K451] and PD-L1 [1:500, Abcam, cat no-ab205921]). A panoramic scanner (3DHISTECH) was used to take the TMA images. Images were analyzed by HistoQuant^TM^ software (3DHISTECH). For both Ki67 and Pd-L1, the number of positive nuclei was calculated, and the value was expressed as object frequency (pcs/mm^2^).

## 3. Results

All five patients had a significant response to combined treatment with pembrolizumab and lenvatinib (Table 1). In addition, patients’ CD8+ T cells’ sensitivity for pembrolizumab after exposure to lenvatinib treatment was explored using radiological and laboratory analyses. Common adverse events (AE) shown in Table 2.

### 3.1. Case Series

#### 3.1.1. Case 1—Excellent Response Combined with Disappearance of the Lower Left Lung after Treatment with Lenvatinib and Pembrolizumab

A 68-year-old man with cough and shortness of breath (patient 1) was referred to the emergency department in November 2018. He was a smoker (30 pack/year over the previous 20 years) receiving treatment for diabetes mellitus type 2, prostatic benign hyperplasia, and hyperlipidemia. There was no family history of cancer. Chest radiography showed a right upper lung (RUL) ground-glass opacity. Upon admittance, a computed tomography (CT) scan of the pleural cavity showed a 2.5-cm mass in the RUL. Positron emission tomography–computed tomography (PET-CT) showed hypermetabolic uptake in the RUL (the 2.5-cm mass) and hypermetabolic uptake in the left adrenal (3.5 cm in diameter), as well as in the retropancreatic region (2 cm). The pathologic stage was determined to be T1C N0 M1 (stage 4C). Brain imaging excluded CNS metastatic disease.

A biopsy was taken under CT guidance, with the histopathology from the RUL mass showing adenocarcinoma of lung origin. A molecular testing panel of the tumor tissue showed the presence of 21 mutations (none of them treatable), including MDM2 (murine double minute 2), KRAS (Kirsten rat sarcoma) amplification, RB1 (retinoblastoma protein) amplification, and STK11 (serine/threonine kinase 11).

The patient underwent left adrenalectomy and, later, right upper lobectomy. The patient received systemic intravenous chemoimmunotherapy of pemetrexed (500 mg/m^2^) plus carboplatin (dosed to AUC-5) and pembrolizumab (at a dosage of 200 mg) on day 1, every 21 d, with a partial response. Pembrolizumab and pemetrexed were continued as maintenance. After eight months of maintenance therapy, a PET-CT in October 2019 showed disease progression with hypermetabolic uptake in a right adrenal mass (4 cm in diameter) as well as a new mass in the right pectoralis (1.4 cm in diameter) and suspected metastasis in the sigmoid colon (2 cm in diameter). A hypermetabolic uptake (3 cm in diameter) was seen in the left lower lung (LLL). The patient received radiotherapy to the right adrenal gland and, one week after concluding radiotherapy treatment, underwent both a right mastectomy and resection of the colon. Both resections showed adenocarcinoma of lung origin.

A month later, the patient received one dose of intravenous docetaxel (at a dosage of 75 mg/m^2^) in the context of a clinical trial. He suffered from severe adverse effects including grade 3 diarrhea, grade 2 neuropathy, and severe weakness, resulting in his leaving the trial after the single dose.

In June 2020, the patient resumed pembrolizumab (at a dosage of 200 mg) on day 1, every 21 days with the addition of daily oral lenvatinib (14 mg). Approximately 40 days later, CT of the chest and abdomen revealed the complete disappearance of the LLL nodule and shrinking of the right adrenal mass in (Figure 1). Currently, (June 2021), the patient is still on pelbrolizumab and lenvatinib 14 mg QD with no evidence of disease on his updated scans.

#### 3.1.2. Case 2—A Multiple Radiological Response Followed by Less Metabolic Uptake Immediately after Combination of Lenvatinib and Pembrolizumab

A 68-year-old man (patient 2) was referred to the emergency department in March 2018 due to fainting and loss of consciousness. He had a history of malignant melanoma (stage 1) resected from the chest wall, was a smoker (60 pack/year over the previous 45 years) and was receiving treatment for hypertension. There was no family history of cancer. CT scan showed RUL two nodules (1.3 cm and 0.8 cm,), and PET-CT showed hypermetabolic uptake in both nodules (MRI) and no evidence of brain metastases. The pathologic stage was determined to be T1b N0 M0 (stage 1B). The patient underwent bi-segmentectomy of the RUL. Histopathologic findings of both nodules were adenocarcinoma of lung origin, with clean margins.

The patient remained disease free until March, 2020, when the PET-CT showed hypermetabolic uptake in the left upper lung (LUL) (2.7 cm × 1.6 cm in diameter); a LLL nodule with a diameter of up to 4 mm, which was too small in size to show uptake on PET; hypermetabolic uptake in the spinal bones (C4, C6); and hypermetabolic uptake in the paraspinal muscle and in two major muscles in the left anterior thigh. There was no evidence of brain metastases. The new pathologic stage was determined to be T2 N0 M1c (stage 4B). The patient received systemic intravenous chemoimmunotherapy consisting of pemetrexed (500 mg/m^2^) plus carboplatin (dosed to AUC-5) and pembrolizumab (at a dosage of 200 mg) on day 1, every 21 days; after two cycles a PET-CT revealed continued disease progression; new findings with hypermetabolic uptake were found in the left hilar lymph node and in the paravertebral area of the spinal bones (D3 and D5). New biopsies were taken from the paravertebral area.

After the fourth cycle, August 2020, disease progression was stable, but only the two metastases in the major muscles of the left anterior thigh were reduced (by 30%). Pathology results were negative for melanoma and positive for lung adenocarcinoma. Pemetrexed was stopped, but the patient continued with pembrolizumab (at a dosage of 200 mg) on day 1, every 21 days with daily oral lenvatinib (at a dosage of 20 mg). A PET-CT from September 2020 showed a significant radiological response in the LUL mass (0.2 cm in diameter) as well as in the metabolic uptake of the left hilar lymph node and in the paravertebral area. In addition, a significant radiological response (diameter and metabolic uptake) of the two major muscles in the left anterior thigh was seen (Figure 2).

Currently, June 2021, he is still on pembrolizumab with lenvatinib 14 mg QD with olgoprogression of D10 lesion that has been radiated stereotactically.

#### 3.1.3. Case 3—Radiological Improvement Achieved after Treatment with Pembrolizumab and Lenvatinib in a Patient after Hospitalization in an Intensive Care Unit

A 64-year-old woman (patient 3) was referred to the emergency department in February 2020 for cough and progressive shortness of breath. She had a history of uterine papillary serous carcinoma (stage 1) in 2011, which had been resected. She was an active smoker (40 pack/year during the previous 28 years) and was receiving treatment for diabetes mellitus type 2.

Chest radiography (CXR) showed RUL mass (diameter of 6 cm) and a right pleural effusion. Draining of the pleural effusion showed adenocarcinoma of lung origin. A PET-CT showed hypermetabolic uptake in the RUL mass (10.6 cm × 10.5 cm in diameter) that was penetrating the mediastinum with right pleural effusion, with several masses connected to the right internal pleura (diameter of 1.5 cm). There was no evidence of brain metastases. The pathologic stage was T4 N2 M1 (stage 4B). The patient received systemic intravenous chemoimmunotherapy consisting of pemetrexed (500 mg/m^2^) plus carboplatin (dosed to AUC-5) and pembrolizumab (at a dosage of 200 mg) on day 1, every 21 days. After three cycles, a chest CT showed stable disease, and after the fifth cycle, PET-CT (May 2020) showed stable disease. Two weeks later, the patient was hospitalized in the intensive care unit owing to oxygen desaturation, sepsis, and acute renal failure. She was treated with artificial respiration and underwent insertion of a stent to the right lung, which opened the airway to the RUL. Four days later, she was extubated. In July 2020, combined treatment with lenvatinib and pembrolizumab began, and after three weeks, a PET-CT showed reduction in the intensity of the absorption in the extensive hypermetabolic findings in the right lung, but hypermetabolic uptake was seen in the eighth and ninth left ribs (suspected fracture) (Figure 3).

#### 3.1.4. Case 4—A Significant Response to Pembrolizumab, Lenvatinib, and Gemcitabine in a Patient with Malignant Pleural Mesothelioma

A 50-year-old woman (patient 4) was referred to the emergency department in January 2018 for chest discomfort and shortness of breath. She was generally healthy with no chronic disease, a smoker (10 pack/year over the previous 15 years) and was receiving treatment for hypertension. Her father had been diagnosed at age 70 with mesothelioma.

CXR showed a left pleural effusion. A drainage of the pleural effusion was performed, and histopathologic findings showed malignant epithelioid pleural mesothelioma. The PET-CT showed hypermetabolic uptake and thickening in the left circumferential pleura and bilateral hypermetabolic uptake and enlargement of lymph nodes (LNs) in the mediastinal area. The pathologic stage was determined to be T4 N2 M0 (stage 3B). The patient received systemic intravenous chemoimmunotherapy therapy consisting of pemetrexed (500 mg/m^2^) plus carboplatin (dosed to AUC-5) and bevacizumab (15 mg/kg) on day 1, every 21 days for five cycles. PET-CT showed good partial response in the thickening of the left circumferential pleura and the mediastinal LNs, and the patient had a left lung decortication. Carboplatin, pemetrexed, and bevacizumab were represcribed at the same dosages; after two cycles, bevacizumab and carboplatin were discontinued due to brain thrombosis, and the patient remained on pemetrexed for another cycle. The patient underwent PET-CT (April 2019) that showed disease progression with hypermetabolic uptake and enlargement of the right mediastinal LN (1.4 cm in diameter), left supraclavicular LN (1.7 cm in diameter), and left axilla LN (2.3 cm in diameter) and a hypermetabolic uptake in several retroperitoneal LN. The treatment was changed to pembrolizumab (at a dosage of 200 mg) on day 1, every 21 days for five cycles. PET-CT showed partial response, and the left axillary LN was resected. Ipilimumab (1 mg/kg every 4 weeks) was added to the pembrolizumab for five cycles; PET-CT showed significant partial response, and ipilimumab was stopped owing to liver injury (immunotherapy-induced grade 2 toxicity. Pembrolizumab was continued alone for five cycles. PET-CT (June 2020) showed disease progression with hypermetabolic uptake in the left pleural hemithorax and spinal bone (D6), and hypermetabolic uptake and enlargement in the left supraclavicular LN, left axillary LN, and retroperitoneal LN. Pembrolizumab was continued with the addition of oral lenvatinib daily (dosage of 20 mg) and gemcitabine (1000 mg/m^2^ on day 1, every 21 days). After two months, PET-CT from August 2020 showed a significant decrease of hypermetabolic uptake and enlargement in the left supraclavicular LN, left axillary LN, left area of the pleural hemithorax, retroperitoneal LN, and spinal bone (D6) (Figure 4). She had local progression and lenvatinib hads been stopped at 12/2020, therefore providing a PFS of nine months. Thereafter, she has continued with pembrolizumab and gemcitabine with good disease control.

#### 3.1.5. Case 5—Combined Treatment Stabilized the Patient’s Widespread Disease for a Period of 10 months

A 61-year-old man (patient 5) was seen in August 2016 for cough and progressive shortness of breath (especially during effort) of three months’ duration. He had a history of hypertension, and he was a smoker (35 pack/year over the previous 20 years), with no family history of cancer. CXR showed a ground-glass opacity in the LUL. Pneumonia was suspected, and he was treated with antibiotics. One month later he underwent follow-up CXR, which still showed the RML ground-glass opacity. A CT scan showed a LUL mass (5.5 cm × 4.6 cm). PET-CT showed hypermetabolic uptake in the LUL mass, hypermetabolic uptake and enlargement of the mediastinal and retroperitoneal LNs, and hypermetabolic uptake in the left femur and left pelvis (up to 1 cm), and a biopsy finding of adenocarcinoma of lung origin. There was no evidence of brain metastases. The tumor tissue was positive for KRAS, STK11, and PDL1 < 24%. The patient received systemic intravenous chemotherapy and vascular endothelial growth factor therapy consisting of pemetrexed (500 mg/m^2^), cisplatin (75 mg/m^2^), and bevacizumab (15 mg/kg), all given on day 1, every 21 days for 5 cycles. In March 2017, treatment was changed to a combination of chemoimmunotherapy pemetrexed (same dosage) and pembrolizumab (at a dosage of 200 mg) on day 1, every 21 days, with a complete metabolic response. Six months later, pemetrexed was stopped and the patient continued treatment with pembrolizumab (same dosage) but stopped in March 2018 due to weakness and fatigue. One month afterward the patient reported improvement of symptoms; however, PET-CT showed a hypermetabolic left hilar and sub pancreatic uptake. Thereafter, pembrolizumab (same dosage) was represcribed for three cycles, but there was further disease progression. The patient received systemic intravenous chemoimmunotherapy consisting of pemetrexed (500 mg/m^2^) plus carboplatin (dosed to AUC-5) and pembrolizumab (at a dosage of 200 mg) on day 1, every 21 days for 4 cycles. In January 2019, mild local progression (in the lung) of the disease was seen, so hypofractionated radiation therapy to the left hilar and to the mediastinum was added (30 Gy in 10 fractions), and treatment with pemetrexed was continued. In May 2019, PET-CT showed progression of the disease in several sites, including the groin, iliac, mediastinum, and neck. Treatment with pembrolizumab and ipilimumab was then initiated for four cycles; this was followed by further progression of the disease, exhibited by PET-CT. The patient went through dissection of the inguinal LN confirmed by pathologic testing to be a TTF1-positive adenocarcinoma lesion. Next, chemotherapy treatment was renewed with cisplatin (75 mg/m^2^) and pemetrexed (500 mg/m^2^). After the fourth cycle, the patient showed stable disease. In January 2020, the patient was treated with pembrolizumab (at a dosage of 200 mg) on day 1, every 21 d and lenvatinib (at a dosage of 20 mg daily). One month later, significant regression of the lesions was seen with improvement in lung findings and a decrease in the diameter of LNs; three months afterward, the mediastinal lesion was decreased by half of its size. A PET-CT from May 2020 showed no further disease progression. At this stage, the lenvatinib dosage was decreased to 14 mg daily because of aggravation of mouth ulcers. The patient continued the combined treatment, which stabilized his disease with no adverse events for a period of 10 months, at which time the patient suffered a sudden deterioration, eventually leading to death (Figure 5).

### 3.2. Testing CD8+ T Cell Responses of Four Patients to Lenvatinib In Vitro

We further studied the importance of our results for human T cell responses on a more mechanistic level. We first withdrew blood from and isolated the PBMCs of patients 1, 2, 4, and 5, using the Ficoll gradient. Two days later, CD8+ T cells were isolated and cultured with 200 U IL-2 for a week. The four patients were clinically negative responders to pembrolizumab. In vitro treatment with anti-PD-1 with or without lenvatinib on A549 (carcinoma cell line) seeded on 96-well plates with A549 showed a significant elevation secretion of IFN-γ by the primary CD8+ T cells of patients 4 and 5 in response to the combined treatments, compared with IFN-γ levels secreted by these cells following treatment with anti-PD1 alone (Figure 6). Fully saturated secretion of IFN-γ was seen in the primary CD8+ T cells of patients 1 and 2 only by using the anti-PD-1 (Figure 6).

### 3.3. Pre-Treatment of Cancer Ex Vivo with Lenvatinib Contribute to Reduction of PD-L1 Expression and to Reduced Proliferation Levels While Using T Cells and Pembrolizumab

We also performed the i-TEVA method using PDX and T cells of patient 4. We examined the outcomes of three main combinations: tumor with media, tumor with T cells, and tumor with T cells and pembrolizumab. Each group was tested with and without pretreatment of tumor with lenvatinib. Then, we stained the FFPE sections either with PD-L1 or Ki67 (proliferation marker) (Figure 7A,B). Figure 7A shows Ki67 staining; the group pre-treated with lenvatinib and then with pembrolizumab and T cells had significantly lower proliferation than the untreated group. Figure 7B shows PD-L1 staining; a significant reduction of PD-L1 expression was observed for the PDX explants co-incubated with T cells and pembrolizumab when these explants were pre-treated with lenvatinib. Figure 7C shows representative pictures of each group.

## 4. Discussion

We have described five patients with metastatic adenocarcinoma of the lung (stage IV on presentation). All five patients had progressed despite receiving various therapies. However, after receiving the combination of immunotherapy and TKI, significant positive responses were observed among all patients. Patients 1, 2, and 5, all of whom had NSCLC, showed a significant radiological reduction followed by a near-complete cessation of widespread metastatic disease shortly after administrating the combination of TKI and immunotherapy treatment. Patient 3, who had aggressive and resistant disease, presented a partial radiological reduction less than a month after initiation of the combined treatment. Patient 4, diagnosed with MPM, also had a significant radiological reduction of the widespread and aggressive metastatic disease.

Our results showed a significant elevation in IFN-γ secretion levels after exposure to anti-PD-1 combined with lenvatinib, compared with anti-PD-1 in patients 4 and 5. However, patients 1 and 2 had fully saturated IFN-γ levels at exposure to anti-PD-1 with no difference after the addition of lenvatinib. A reasonable explanation to these findings might be the patient cell status. Patients 1 and 2 were under clinical treatment of lenvatinib and pembrolizumab according to their treatment scheme, whereas patient 4 was 4 months after treatment and patient 5 was before clinical treatment with lenvatinib and pembrolizumab. These differences may explain and support our results; CD8+ T cells may have responded rapidly to pembrolizumab after exposure to lenvatinib, signifying that the CD8+ T cells’ sensitivity to pembrolizumab after treatment with lenvatinib was improved at least in the short term. We used a unique assay to investigate the role of lenvatinib in combination with pembrolizumab (Figure 7). It is known that PD-L1 acts as an immune checkpoint in many solid tumors, including NSCLC [14,15,16,17,18,19,20]; it also known that many tumors successfully overcome the effect of any treatment by generating a new resistant mutation [21,22,23,24]. We did see confirmation of both occurrences in our analyses, but interestingly, once the tumor was pretreated with lenvatinib, lenvatinib seemed to block the PD-L1 upregulation levels, thus making the tumor visible for the immune system.

Our study was limited by the small number of participants. In addition, the response to lenvatinib alone was not tested in this study. Therefore, the responses observed may possibly be a result of the lenvatinib treatment and not necessarily of the combination with pembrolizumab. Nonetheless, promising preclinical data from recent ongoing trials have shown robust antitumor activity with a manageable safety profile in multiple solid tumors with limited treatment options, including lung cancer [25]. We intend to conduct larger scale studies to address these current limitations.

## 5. Conclusions

According to our individual patient observation data and the in vitro study, treatment with lenvatinib and pembrolizumab may provide a beneficial treatment option for patients with NSCLC and MPM, as previously suggested by recent clinical trials.

We hope to further explore the combination of anti-PD-1 therapy and multikinase targeted therapy for different carcinoma treatments in the future.

## Figures and Tables

**Figure 1 cancers-13-03630-f001:**
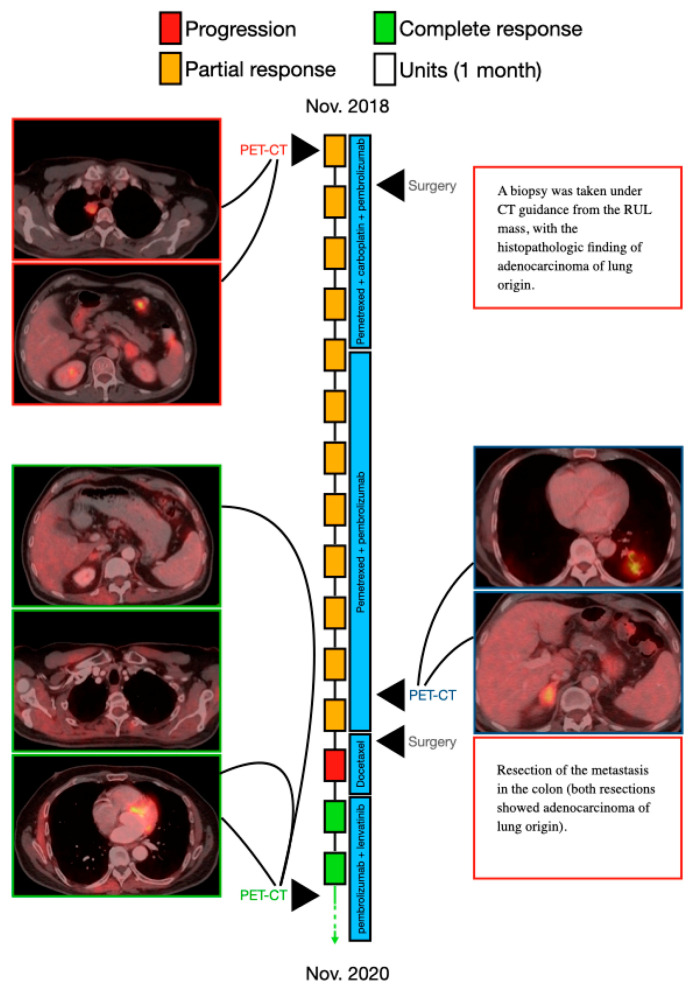
A schematic illustration of patient 1′s timeline, between November 2018 (diagnosis of the disease) and November 2020, including the course of treatment and radiological follow-up. Each rectangle represents 1 month, and the response to treatment during each month is color-coded in green, red, or yellow, as described in the upper legend. Treatment administered at each timepoint is indicated by the blue rectangles. Radiological findings and additional landmarks are presented at relevant points throughout the timeline.

**Figure 2 cancers-13-03630-f002:**
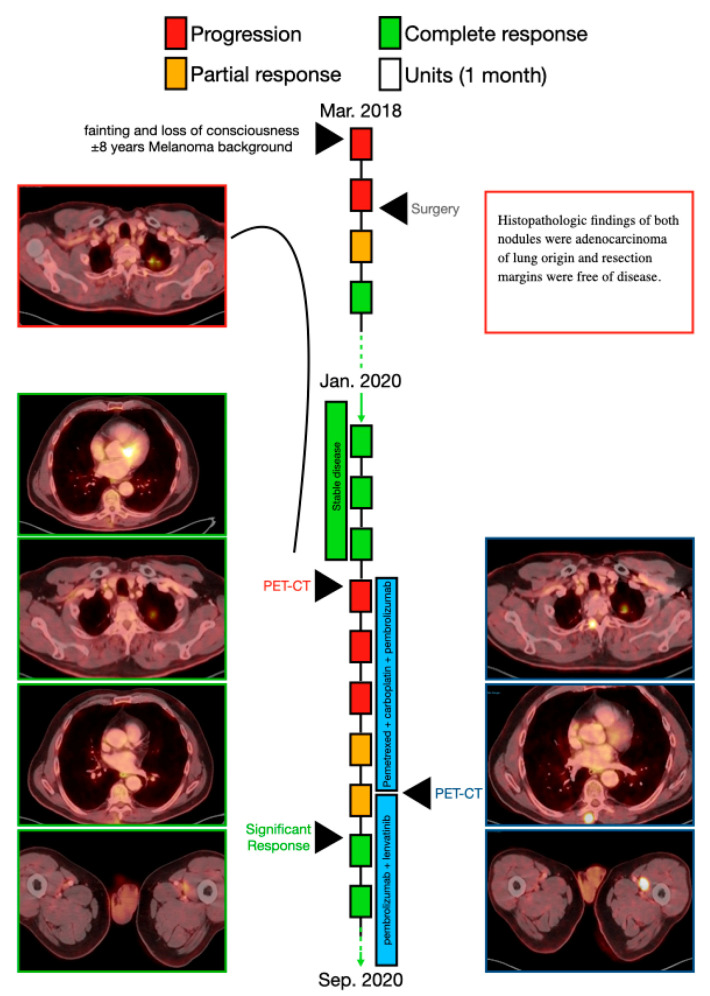
A schematic illustration of patient 2′s timeline, between March 2018 (diagnosis of the disease) and September 2020, including the course of treatment and radiological follow-up. Each rectangle represents 1 month, and the response to treatment during each month is color-coded in green, red, or yellow, as described in the upper legend. Treatment administered in each timepoint is indicated by the blue rectangles. Radiological findings and additional landmarks are presented at relevant points throughout the timeline.

**Figure 3 cancers-13-03630-f003:**
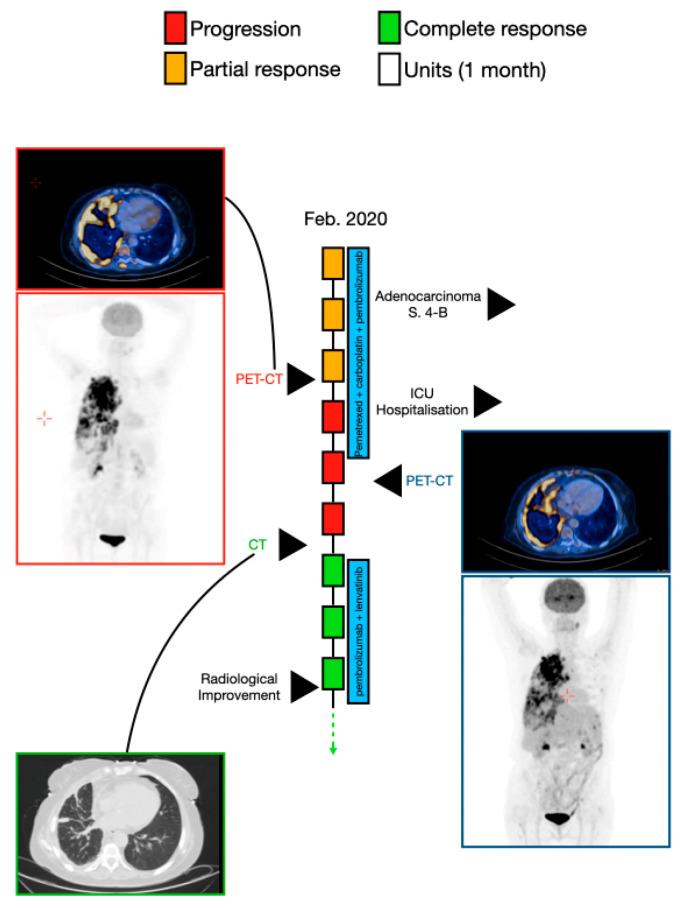
A schematic illustration of patient 3′s timeline, between February 2020 (diagnosis of the disease) and November 2020, including the course of treatment and radiological follow-up. Each rectangle represents 1 month, and the response to treatment during each month is color-coded in green, red, or yellow, as described in the upper legend. Treatment administered in each timepoint is indicated by the blue rectangles. Radiological findings and additional landmarks are presented at relevant points throughout the timeline.

**Figure 4 cancers-13-03630-f004:**
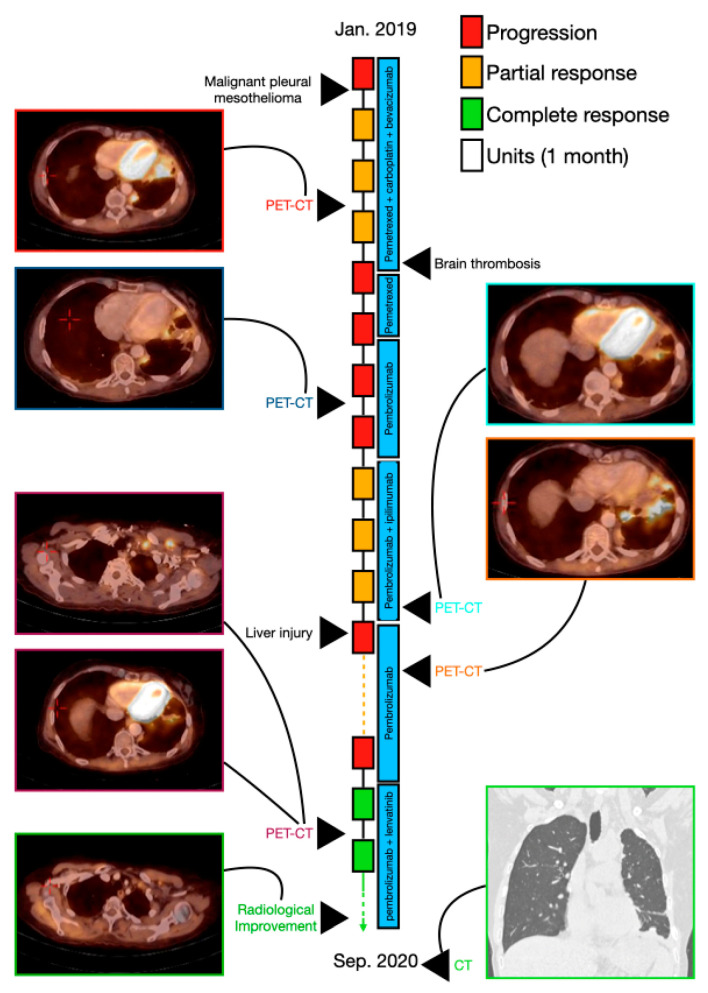
A schematic illustration of patient 4′s timeline, between January 2019 (diagnosis of the disease) and September 2020, including the course of treatment and radiological follow-up. Each rectangle represents 1 month, and the response to treatment during each month is color-coded in green, red, or yellow, as described in the upper legend. Treatment administered in each timepoint is indicated by the blue rectangles. Radiological findings and additional landmarks are presented at relevant points throughout the timeline.

**Figure 5 cancers-13-03630-f005:**
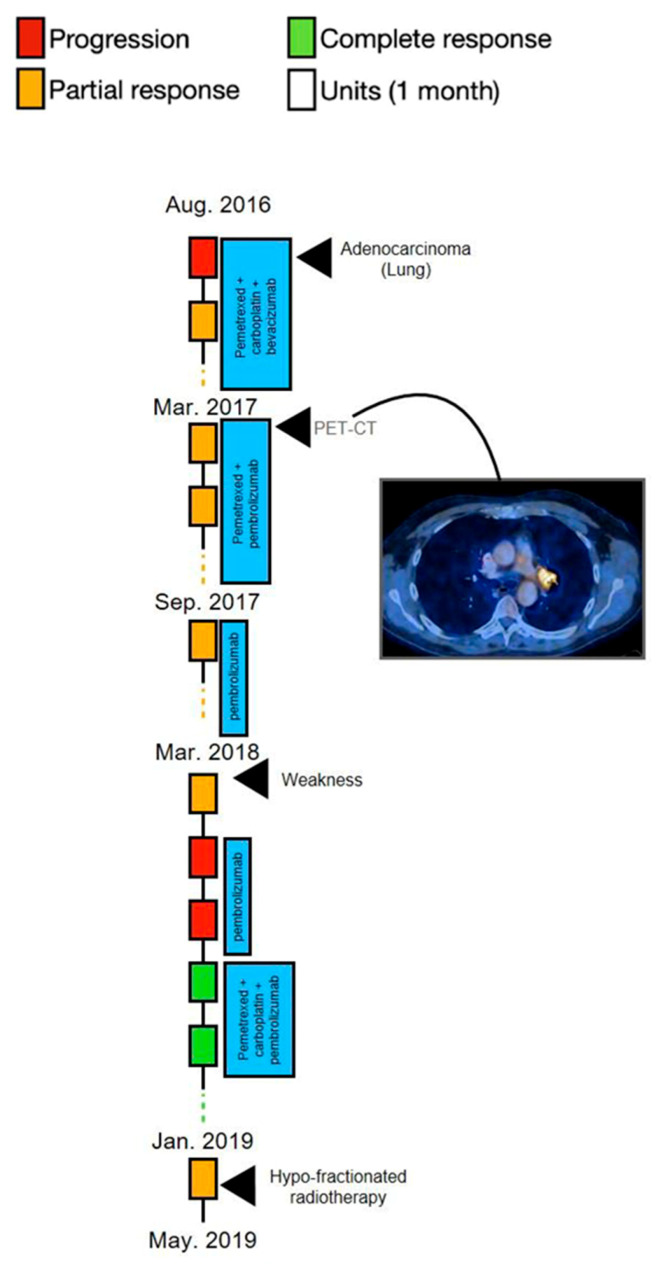
A schematic illustration of patient 5′s timeline, between August 2016 (diagnosis of the disease) and September 2020 (patient’s death), including the course of treatment and radiological follow-up. Each rectangle represents 1 month, and the response to treatment during each month is color-coded in green, red, or yellow, as described in the upper legend. Treatment administered in each timepoint is indicated by the blue rectangles. Radiological findings and additional landmarks are presented at relevant points throughout the timeline.

**Figure 6 cancers-13-03630-f006:**
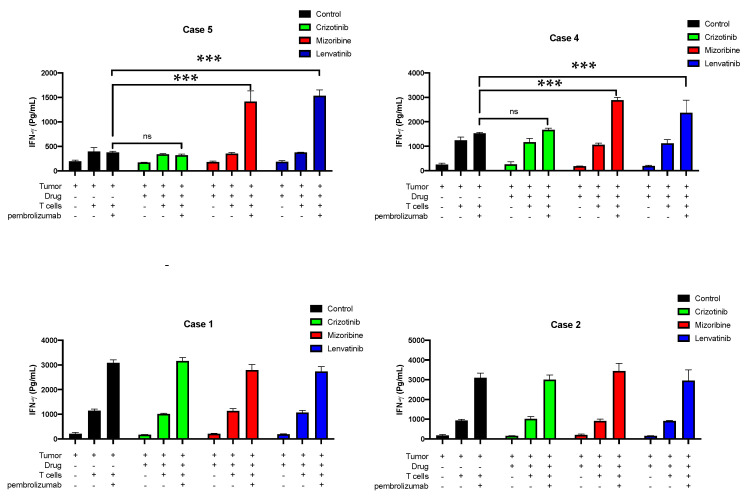
In vitro treatment with anti-PD-1 with or without lenvatinib for CD8+ T cells of patients 1, 2, 4, and 5. All four patients were tested in vitro for 12 different conditions in triplicates. This figure shows the IFN-γ values in pg units upon 12 different conditions. mizoribine (red) and crizotinib (green) were used as positive and negative controls, respectively; the control (black) was untreated, and lenvatinib (blue) was the experimental test. A549 cells were first seeded on 96-well plates 24 h before adding the drugs. Then, drugs were added, and cells were incubated for 10 h. Media with or without anti-PD-1 antibody (pembrolizumab) containing the T cells of different patients was added after performing three PBSX1 washes for the whole 96-well plate. Human IFN-γ was performed using the collected media from the experimental wells and measured using the enzyme-linked immunosorbent assay (ELISA) method at a wavelength of 650 nm. A two-way analysis of variance (ANOVA) multiple comparisons test was performed among the groups for each patient individually, with three biologically independent counts in each group. *** *p* < 0.0001 for both the mizoribine- and the lenvatinib-treated groups in patients 4 and 5. All statistical data analysis was performed using GraphPad Prism, version 8.0.2.

**Figure 7 cancers-13-03630-f007:**
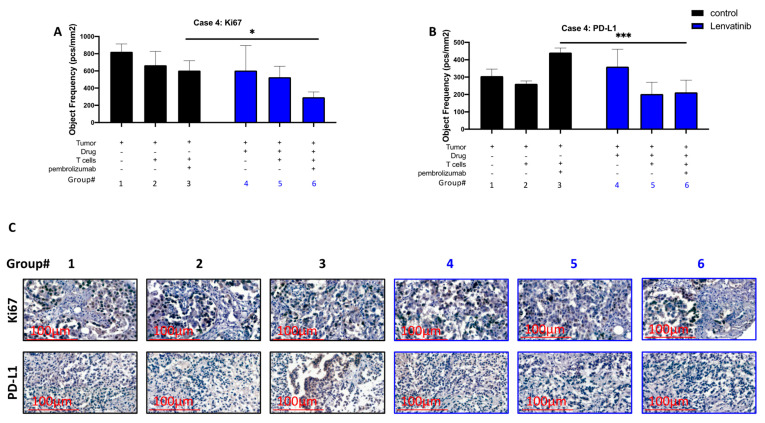
Correlation between the multikinase inhibitor (lenvatinib) and the immune checkpoint inhibitor (pembrolizumab) treatments. A 3-dimensional (3D) ex vivo assay was performed as an i-TEVA assay on patient 4′s tumor and T cells: (**A**,**B**) lenvatinib (blue) was the experimental test, and the control (black) was untreated. From the 3D ex vivo assay, 5-µm FFPE sections were stained either for Ki67 (proliferation marker) or PD-L1 expression. A two-way ANOVA multiple comparisons test was performed among the groups for each patient individually, with three biologically independent counts in each group. * *p* = 0.0498; *** *p* = 0.0007. All statistical data analysis was performed using GraphPad Prism, version 8.0.2; (**C**) This figure shows representative pictures for each group from 1 to 6. Pictures were captured using CaseViewer 40× magnification at 300 dpi; scale bar is located at the left bottom corner, 100 μm (red).

**Table 1 cancers-13-03630-t001:** Summary of clinical characteristics and treatment outcomes of patients presented in cases 1–5.

Case No.	Gender	Age	Smoking History	Histopathologic Diagnosis	Stage at Diagnosis	Genetic Alterations	Treatment before LEN+PEM			LEN + PEM Outcome
							chemotherapy	immunotherapy	VEGF-A	
1	M	68	30 PY	adenocarcinoma of lung	4-C	MDM2, KRAS amplification, RB1 amplification, STK11	13 mo.	12 mo. (combined with chemo		CR after 2 mo +
2	M	68	60 PY	adenocarcinoma of lung	1-B		4 mo.	4 mo. (combined with chemo)		Significant response after 2 mo +
3	F	64	40 PY	adenocarcinoma of lung	4-B		4 mo.	4 mo (combined with chemo)		CR after 3 mo +
4	F	50	10 PY	malignant pleural mesothelioma	3-B		6 mo.	9 mo	4 mo. (combined with chemo)	Significant response after 2 mo. +
5	M	61	35 PY	adenocarcinoma of lung	4-C	KRAS, STK11, PDL1 < 24%	19 mo.	14 mo. (combined with chemo) + 11 mo. (without chemo)	5 mo. (combined with chemo)	Stabilized disease with no adverse events for 10 mo., until patient’s sudden death.

**Table 2 cancers-13-03630-t002:** Common adverse events (AE) seen in all five presented patients following treatment with lenvatinib plus pembrolizumab. The numbers indicate the grade of each AE in each specific patient.

Case No.	Diarrhea	Fatigue	Hypothyroidism	Hypertension	Weight Loss
1	2	1	1	0	0
2	2	1	0	1	1
3	0	2	0	1	1
4	0	1	0	0	1
5	0	1	1	1	0

## Data Availability

Data is contained within the article or are available from the authors upon reasonable request.

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
