# Peer review of "Rapid Response to the Combination of Lenvatinib and Pembrolizumab in Patients with Advanced Carcinomas (Lung Adenocarcinoma and Malignant Pleural Mesothelioma)"

_cancers, 2021, doi:10.3390/cancers13143630_

Round 1

Reviewer 1 Report

I have no further comments

Reviewer 2 Report

Thank you, my concerns were addressed and I have no further comments

This manuscript is a resubmission of an earlier submission. The following is a list of the peer review reports and author responses from that submission.

Round 1

Reviewer 1 Report

This is an interesting paper demonstrating preliminary responses in 5 patients when combining a traditional anti-PD1 checkpoint with TKI therapy. It is well written, straight to the point and easy to follow.  I have a few comments only.

1) There is no information in the method section about patient selection, consent, treatment description etc. Was it a trial? Is it still ongoing? Where were the patient recruited and was there regulatory and ethical permits?

2) Please improve the figure legend for Fig 6 so it includes more detailed information about the experiment and specifically statistics used and meaning of the different *, *** etc.

3) Please summarize some of the patient characteristics and outcomes in a table so it is easy to compare side by side.

4) A table with AEs would be appreciated as well for easy comparison.

Reviewer 2 Report

Authors made a lot of effort to clearly present the history of patients. All cases were rather non-responders to pembrolizumab containing regimens. The addtion of lenvatinib to pameborlizumab produced responses, but it is impossible to conclude whether the response was because of the combination or because of lenvatinib alone. The conclusion "Combination immunotherapy and TKI is a potentially very effective therapeutic option the new era of biologic therapy for NSCLC" is missleading, as lenvatinib monotherapy could have produced similar responses. The in vitro study is interesting and I would encourage the authors to expand these data that will offer important insights in the mechanisms of activity of  such combinational therapies.